# Residential Surrounding Greenspace and Mental Health in Three Spanish Areas

**DOI:** 10.3390/ijerph17165670

**Published:** 2020-08-05

**Authors:** Maria Torres Toda, Asier Anabitarte Riol, Marta Cirach, Marisa Estarlich, Ana Fernández-Somoano, Llúcia González-Safont, Mònica Guxens, Jordi Julvez, Isolina Riaño-Galán, Jordi Sunyer, Payam Dadvand

**Affiliations:** 1Barcelona Institute for Global Health (ISGlobal), 08003 Barcelona, Spain; maria.torres@isglobal.org (M.T.T.); marta.cirach@isglobal.org (M.C.); monica.guxens@isglobal.org (M.G.); jordi.julvez@isglobal.org (J.J.); jordi.sunyer@isglobal.org (J.S.); 2Campus del Mar, Pompeu Fabra University (UPF), 08003 Barcelona, Spain; 3Spanish Consortium for Research on Epidemiology and Public Health (CIBERESP), 28029 Madrid, Spain; estarlich_mar@gva.es (M.E.); fernandezsana@uniovi.es (A.F.-S.); gonzalez_llu@gva.es (L.G.-S.); isogalan@yahoo.es (I.R.-G.); 4Biodonostia Health Research Institute, Environmental Epidemiology and Child Development Group, 20014 San Sebastian, Spain; a-anabitarteriol@euskadi.eus; 5University of the Basque Country (UPV/EHU), Preventative Medicine and Public Health Department, Faculty of Medicine, 48940 Leioa, Spain; 6Epidemiology and Environmental Health Joint Research Unit, FISABIO-Universitat Jaume I-Universitat de València, 46010 Valencia, Spain; 7Department of Infirmary and Chiropody, Universitat de València, 46010 Valencia, Spain; 8Unit of Molecular Cancer Epidemiology, University Institute of Oncology of the Principality of Asturias (IUOPA) – Department of Medicine, University of Oviedo, 33006 Oviedo, Spain; 9Instituto de Investigación Sanitaria del Principado de Asturias (ISPA), 33011 Oviedo, Spain; 10Department of Child and Adolescent Psychiatry/Psychology, Erasmus University Medical Centre–Sophia Children’s Hospital, 3015 Rotterdam, The Netherlands; 11Institut d’Investigació Sanitària Pere Virgili (IISPV), Hospital Universitari Sant Joan de Reus, 43204 Reus, Spain; 12Servicio de Pediatría. Endocrinología, Hospital Universitario Central de Asturias (HUCA), 33011 Oviedo, Spain; 13Municipal Institute of Medical Research (IMIM-Hospital del Mar), 08003 Barcelona, Spain

**Keywords:** nature, mental illness, psychiatric disorder, psychosomatic symptoms, parks

## Abstract

Exposure to greenspace has been related to improved mental health, but the available evidence is limited and findings are heterogeneous across different areas. We aimed to evaluate the associations between residential exposure to greenspace and specific psychopathological and psychosomatic symptoms related to mental health among mothers from a Spanish birth cohort. Our study was based on data from 1171 women participating in two follow-ups of a population-based cohort in Valencia, Sabadell, and Gipuzkoa (2004–2012). For each participant, residential surrounding greenspace was estimated as the average of the satellite-based Normalized Difference Vegetation Index (NDVI) across different buffers around the residential address at the time of delivery and at the 4-year follow-up. The Symptom Checklist 90 Revised (SCL-90-R) was applied to characterize mental health at the 4-year follow-up. We developed mixed-effects logistic regression models controlled for relevant covariates to evaluate the associations. Higher residential surrounding greenspace was associated with a lower risk of somatization and anxiety symptoms. For General Severity Index (GSI), obsessive–compulsive, interpersonal sensitivity, depression, hostility, phobic anxiety, paranoid ideation, and psychoticism symptoms, we generally observed protective associations, but none attained statistical significance. Findings from this study suggested a potential positive impactof greenspace on mental health.

## 1. Introduction

Nowadays, mental disorders are a major cause of non-fatal global burden of disease (GBD, [1]). Around one in seven people globally (11–18%) has one or more mental disorders [1], and this proportion is projected to increase in the coming years [2]. Among mental disorders, anxiety and depression are the most common disorders [3]. Depression alone is the leading cause of mental health-related disease burden, affecting approximately 300 million people worldwide [4]. The majority of them are women who are twice as likely to develop depression and anxiety than men [5]. A new Lancet Commission report on mental health reported a rise of mental disorders in every country in the world, resulting in a global economic cost of $16 trillion by 2030 [6]. In Europe, it has been estimated that mental disorders, particularly depression and anxiety, affect more than a third of the population every year [7]. In Spain, psychiatric disorders, including mental disorders, are estimated to impose an economic impact equivalent to almost 8% of the country’s Gross Domestic Product (GDP) [8]. The promotion of mental health and the prevention of mental disorders are fundamental to improvethe quality of life, well-being, and productivity of individuals and communities.

By the year 2050, 68% of the world’s population is projected to live in urban areas (World Urbanization Prospects [9]), where there is a higher prevalence of mental disorders compared to rural areas [10]. Urban residents often have limited access to natural environments, including green spaces, while contact with green spaces has been associated with improved mental health [11,12,13]. Earlier studies were mainly experimental, looking at the short-term beneficial impactsof contact with greenspace on mental health [14]. More recently, a growing body of epidemiological evidence has supported a beneficial impact of long-term exposure to greenspace on mental health and well-being [11,12,13,15,16]. A study in Barcelona, for example, found that increasing residential surrounding greenspace was associated with self-reported history of depression [Odds Ratio (OR) (95% Confidence Interval (CI)) = 0.18 (0.06, 0.58)] [11]. Additionally, in the United Kingdom (UK), Sarkaret al. [15] found a protective association of greenspace with depression [OR (95% CI)= 0.96 (0.93, 0.99)] in a sample of 122,993 adults. However, evidence has remained limited on this aspect [16], and the reported findings are heterogeneous across different areas [17].

Several mechanisms have been proposed to explain how exposure to greenspace may influence mental health. Among these proposed mechanisms, mental restoration and stress reduction are considered to be among the most relevant ones [16,18]. Other potential mechanisms include the enhanced social contact and sense of community, increased physical activity, and reduced exposure to air pollution, noise, and heat [16,19].

Using information from a well-established Spanish birth cohort across three areas, we aimed to fill gaps in the current literature by evaluating whether residential exposure to greenspace was associated with mental health in adult women. This study relied on a broad range of psychopathological and psychosomatic symptoms related to mental health together to provide a comprehensive perspective over the aforementioned association.

## 2. Materials and Methods

### 2.1. Study Population

Our study was based on data obtained from mothers participating in a population-based birth cohort, INMA (INfancia y MedioAmbiente; Environment and Childhood), across three areas in Spain (Figure 1). These three areas, namely, Valencia, Sabadell, and Gipuzkoa, are located in the eastern, northeastern, and northern parts of Spain, respectively (Figure 1). Valencia and Sabadell are part of the Mediterranean region, characterized by a dry climate with hot and dry summers, mild winters, and maximum vegetation in autumn and spring [20]. Eurosiberian region, where Gipuzkoa is included, is characterized by a humid and windy climate with precipitation throughout the year, relatively cold winters, and maximum vegetation during summer months [20].

Pregnant women in their first trimester of pregnancy were enrolled in the cohort based on the following inclusion criteria (i) being resident in one of the study areas, (ii) being at least 16 years old, (iii) having a singleton pregnancy, (iv) not having used assisted reproductive techniques, (v) willing to deliver in the reference hospital, and (vi) having no communication problems. All participants provided written informed consent before enrollment to the study and the INMA project was approved by the ethics committee in each area. All the data used in the present study were collected between 2004 and 2012 (see Appendix A). Additionally, additional information on data collection and on INMA cohort has already been published elsewhere [21]. The current study was based on the maternal residential addresses at the time of delivery as well as the 4-year follow up to assess exposure to greenspace and the evaluation of the maternal mental health at the 4-year follow up.

### 2.2. Assessment of Residential Surrounding Greenspace

Residential surrounding greenspace was assessed using the Normalized Difference Vegetation Index (NDVI) (Landsat 4–5 TM data at 30 m × 30 m resolution). NDVI is a commonly used indicator of greenspace obtained from satellite imagery based on the land surface reflectance of visible (red) and near-infrared parts of the spectrum. Values of NDVI range from −1 to 1, with higher values indicating more photosynthetically active vegetation cover [22]. Sattelite images were selected for each cohort within the greenest months and clear-sky (cloud-free) conditions. Satellite imagery was atmospherically corrected and converted to NDVI (Figure 2). For each participant, we estimated residential surrounding greenspace as the average of NDVI within buffers of 100 m, 300 m, and 500 m [23] around the geocoded residential address. We assessed residential surrounding greenspace at two time points: at the time of delivery and at the 4-year follow-up. For the main analyses, we used the average of each buffer of greenspace levels over these two time points to obtain the residential surrounding greenspace for the four years prior to the assessment of mental health. For the sensitivity analyses, we used only residential surrounding greenspace at the 4-year follow-up to consider greenspace levels at the same time when mental health was assessed.

### 2.3. Assessment of Mental Health

To characterize the mental health in our participants, we applied the Symptom Checklist 90 Revised (SCL-90-R), which is a self-reported questionnaire widely used to assess mental health. We applied the Spanish version of this questionnaire [24], and women filled it in at the 4-year follow-up. The SCL-90-R comprises 90 items, each describing specific psychopathological or psychosomatic symptoms. Items are evaluated on a 4-point scale ranging from 0 (absence of the symptom) to 4 (maximum discomfort). The participants chose a grade (0–4) for each item that corresponded to how they had felt over the previous seven days [24]. Completing the questionnaire required approximately 20 min.

The 90 items are grouped into nine symptomatic dimensions classified as somatization, obsessive–compulsive, interpersonal sensitivity, depression, anxiety, hostility, phobic anxiety, paranoid ideation and psychoticism. Each dimension is assessed with between 6 and 13 items, and higher scores reflect more severity problems. The questionnaire also provides a general severity index (GSI), which is a mean score of all 90 items. For our analyses, we converted raw scores into T-scores. Then, according to the test guidelines [24], we dichotomized T-scores by considering T-scores more than or equal to 65 as being at risk and T-scores less than 65 as not being at risk. We developed dichotomized T-scores separately for each of nine domains as well as for GSI.

### 2.4. Statistical Analyses

To account for the multi-level structure of our data (participants within areas), we developed mixed-effects logistic regression models with dichotomized domain and T-scores as the outcome (one at a time), cohort area as the random effect and residential surrounding greenspace (separately for each buffer) as a fixed-effect predictor. The analyses were further adjusted for age, tobacco smoke (yes/no), IQ (Wechsler Adult Intelligence Scale (WAIS-IV)) and indicators of socioeconomic status (SES) at individual and neighborhood levels. We used educational attainment (primary school/secondary school/university) as the indicator of individual SES and Urban Vulnerability Index as a measure of neighborhood-level of SES. Urban Vulnerability Index was measured at the census tract corresponding to the maternal residential address and was based on 21 indicators of urban vulnerability grouped into four themes: sociodemographic vulnerability (five indicators), socioeconomic vulnerability (six indicators), housing vulnerability (five indicators) and subjective perception of vulnerability (five indicators) [25]. To generate comparable results for different buffers, we reported the association for each interquartile range (IQR) increase in residential surrounding greenspace in each buffer. STATA 14.0 (StataCorp. 2015. *Stata Statistical Software: Release 14*. College Station, TX: StataCorp LP) statistical software was applied to conduct all of our analyses. The level of statistical significance was set at *p* < 0.05.

### 2.5. Sensitivity Analyses

#### 2.5.1. Further Adjustment by Alcohol Consumption and Tobacco Exposure

We further adjusted our analyses for alcohol consumption during pregnancy (yes/no), smoking during pregnancy (yes/no), and second-hand smoking at home (yes/no).

#### 2.5.2. Exclusion of Single-Parent or Non-White Participants

We were not able to adjust our main analyses for marital status or ethnicity due to the small number of participants in single-parent and non-white categories. We therefore conducted sensitivity analyses by excluding single parent and non-white participants.

#### 2.5.3. Development of Models Using Residential Surrounding Greenspace at the 4-Year Follow-Up

For the main analyses, we averaged residential surrounding greenspace at the time of delivery and 4-year follow-up. As sensitivity analyses, we developed a separate set of models using only residential surrounding greenspace at the 4-year follow-up as an alternative set of exposures. We did so in order to evaluate the amount of residential greenspace when the participants completed the SCL-90-R.

### 2.6. Stratified Analyses

We stratified our analyses based on age (less than 35 years old/more or equal than 35 years old) and educational attainment (primary school/high school/university) to assess whether the associations between greenspace exposure and mental health differed across strata of age and SES.

## 3. Results

### 3.1. Study Population Characteristics

In total, 2270 female participants were enrolled in the cohort across three areas, of whom 1171 (444 participants from Valencia, 475 from Sabadell and 252 from Gipuzkoa) were included in this current study. The inclusion/exclusion of the participants was based on availability of the data of residential exposure to greenspace and completion of the SCL-90-R. Appendix A shows differences between excluded and included participants. Table 1 presents the main characteristics of study participants separately by study area.

### 3.2. Greenspace Exposure

As expected, the amount of residential surrounding greenspace in the cohort located in the Eurosiberian region (Gipuzkoa) was higher compared to those cohorts (Valencia and Sabadell) in the Mediterranean region. A detailed description of the estimated residential surrounding greenspace measures has been presented in Table 2. There were strong correlations among averaged residential surrounding greenspace at the time of delivery and at the 4-year follow-up in each buffer (100 m, 300 m, and 500 m) (Spearman’s correlation coefficient ranging between 0.7–0.9). Moreover, there were strong correlations among residential surrounding greenspace in each buffer and in each follow-up, separately (Spearman’s correlation coefficient ranging between 0.8–0.9).

### 3.3. Mental Health

Table 2 presents the results of the SCL-90-R domains and GSI, separately for each study area. Overall, the results were quite similar in the three areas. However, somatization and depressive symptoms were statistically significantly worse among Sabadell participants compared to Valencia and Gipuzkoa participants.

### 3.4. Greenspace and Mental Health

We observed inverse associations between the residential surrounding greenspace and self-reported somatization and anxiety symptoms (Table 3). A 1-IQR increase in the residential surrounding greenspace across buffers of 100 m, 300 m, and 500 m was respectively associated with odds ratio [OR (95% confidence intervals (CIs))] of 0.63 (0.44, 0.90), 0.64 (0.43,0.93) 0.63 (0.43, 0.93) for the self-reported somatization. Moreover, a 1-IQR increase in NDVI across 500m buffer was associated with reduced odds [0.67 (0.45, 0.99)] of anxiety. We did not find any statistically significant association for other symptomatic dimensions of the SCL-90-R (GSI, obsessive-compulsive, interpersonal sensitivity, depression, hostility, phobic anxiety, paranoid ideation, and psychoticism).

### 3.5. Sensitivity Analyses

Further adjustment of our models for alcohol consumption during pregnancy, smoking during pregnancy and second-hand smoking at home did not result in a notable change in our findings (Appendix A). Similarly, our observed associations were consistent with those of the main analyses after excluding non-white participants or single parents (Appendix A). Moreover, we did not observe any considerable change in our findings after using residential surrounding greenspace at 4-year follow-up as the exposure (Appendix A).

### 3.6. Stratified Analyses

After stratifying of our analyses, we did not observe any notable variation in the strength and direction of the associations between residential surrounding greenspace and mental health across strata of participants’ age and maternal educational attainment (Appendix A).

## 4. Discussion

To our knowledge, this is one of the first studies that simultaneously evaluated the association of residential exposure to greenspace with a comprehensive set of psychopathological and psychosomatic symptoms. This study was based on a well-established cohort located in three areas across two distinct biogeographic regions within the Iberian Peninsula. We used a widely used tool (SCL-90-R) to assess mental health and a satellite-derived index of greenspace to estimate the residential surrounding greenspace of each participant at two-time points. We found protective associations between residential surrounding greenspace and somatization and anxiety dimensions of the SCL-90-R. The other dimensions of the SCL-90-R (GSI, obsessive-compulsive, interpersonal sensitivity, depression, hostility, phobic anxiety, paranoid ideation, and psychoticism) were mostly inversely associated with residential surrounding greenspace, though the associations did not attain statistical significance. We did not observe any indication for differences in the associations across strata of participants’ age and educational attainment.

### 4.1. Interpretation of Results in the Context of Available Evidence

Our findings are in line with several previous studies, which showed beneficial associations between residential exposure to greenspace in adults and self-perceived mental health [11,13,26,27,28,29,30,31,32,33]. However, the majority of previous studies are focused on anxiety and depression [11,15,27,30,33,34]. Although our observed association between residential surrounding greenspace and depression did not attain statistical significance, a recent study in South Korea (n=65,128) found lower rates of depressive symptoms among participants living in greener areas [33]. In the United Kingdom, a cross-sectional study [15] reported a protective association between exposure to greenspace and lower risk of major depressive disorder with more benefits among women, participants younger than 60 years, and participants residing in areas with low neighborhood SES or high urbanity. Another study in Miami (USA) also reported a reduced risk of depression by 28% for the participants with the highest greenspace exposure with stronger associations for those living in low-income neighborhoods [27]. A study in Barcelona (Spain) [11] found associations between a 1-IQR increase in NDVI 300m buffer and reduced odds of self-reported anxiety [0.62 (0.43, 0.89)]. The same study found associations between residential access to a major green space and self-reported history of depression [0.18 (0.06, 0.58)] [11]. In Plovdiv (Bulgaria) [30], a study among 529 participants showed reduced anxiety and depressive symptoms in the participants with higher exposure to residential greenspace across different buffer sizes using NDVI and tree cover as green indicators.

We observed a lower risk of somatization symptoms associated with higher residential surrounding greenspace. We are aware of only one relevant study to this outcome, which was published in 2017 [35] and included several indicators of mental wellness (psychological wellbeing, sleep quality, vitality and lack of somatizations) as their outcomes of interest. They observed a direct association between residential surrounding greenspace exposure (buffer of NDVI around participants’ residence) and lack of somatizations; however, the association was not statistically significant. In that study, the lack of somatizations was self-reported by the participants using seven questions adapted from the 4-Dimensional Symptom Questionnaire (4DSQ). However, in our study, somatization dimension in the SCL-90-R was comprised by the following twelve items: headache, dizziness, heartache, backache, nausea, stomachache, sickness, painful muscles, difficulty breathing, shivers, tingling, numbness, throat lump, body weakness and pain in arms or legs [24]. It seems that backache is one of the somatic symptoms associated with somatizations. For example, in 2009, Maas et al. [36] associated lower prevalence of back complaints in participants living in greener environments. We are not aware of any other study reporting on other psychopathological symptoms included in the SCL-90-R questionnaire (i.e.,obsessive-compulsive, interpersonal sensitivity, hostility, phobic anxiety, paranoid ideation and psychoticism). Therefore, it is not possible to compare some of our findings with those of previous studies.

The stratification of our analyses by participants’ age and educational attainment did not show any notable variation in the associations. However, other studies found differences once they stratified by age or by educational attainment. Bos et al. [34] stratified their analyses into six age groups and the largest effect sizes were observed for youngest (18–24 years old) and oldest (more or equal of 65 years old) women. The Positive Health Effects of the Natural Outdoor environment in Typical Populations of different regions in Europe (PHENOTYPE) project aimed to investigate some of the mechanisms underlying the association between exposure to natural outdoor environments and health across four European cities: Barcelona (Spain), Stoke-on-Trent (United Kingdom), Doetinchem (The Netherlands) and Kaunas (Lithuania). They found associations between exposure to greenspace and mental health, which were stronger for males, younger people, those with low-middle education and residents in Doetinchem (The Netherlands) [35]. Additionally, a study in the United Kingdom [15] found stronger beneficial associations between residential surrounding greenspace and depression among women. The results of previous studies stratifying their models by educational attainment were varied across different areas but were suggestive for more benefits among those participants with low and middle educational attainments [35,37]. There is therefore a need to identify the reasons(s) behind these variationsin the different settings with diverse SES, demography and climate to shed lights on potential underlying pathways.

### 4.2. Potential Underlying Mechanisms

The potential mechanisms underlying the beneficial association between exposure to greenspace and mental health are yet to be established; however, reduced stress, increased mental restoration, enhanced social contacts, increased physical activity and reduced exposure to air pollution, heat and noise are suggested to be involved [16,30,32]. The stress restoration theory suggests that greenspace could promote recovery from stress and help to lessen states of arousal and negative thoughts [16,38]. In this context, spending time and being exposed to natural environments can reduce stress [39], which, in turn, could partially explain our observed protective associations between residential surrounding greenspace and anxiety symptoms. Other studies explored physical activity as a protector against somatization symptoms [40], while physical activity itself has been suggested, although inconsistently, to be a mechanism underlying the health benefits of greenspace exposure [12,16]. Previous literature has also associated exposure to higher levels of air pollution with worse mental health [41], but more studies are needed to disentangle the role of air pollution from the association of exposure to greenspace and improved mental health.

### 4.3. Limitations

First of all, this cross-sectional study, by design, had a limited capability to establish causality. Our use of a satellite-based index of vegetation to abstract the residential surrounding greenspace allowed us to characterize all vegetation (even small patches of greenspace) in a standardized way. However, NDVI could not distinguish different types of greenspace, which could have differentially influenced our findings. Similarly, our assessment of greenspace exposure did not take account of the quality of greenspace. Quality aspects such as safety, aesthetics, amenities and level of maintenances of the area might be relevant for our evaluated outcomes and hence lack of including them in our analyses might have affected our findings. For those participants who have changed their residential address, we did not have the date at which they moved, and this might have influenced our exposure assessment. We did not have enough statistical power to evaluate pathologic categories (T-score more or equal to 80) instead of risk categories. Additionally, we did not have data on postnatal depression, which could be a confounding factor in the reported association.

## 5. Conclusions

We observed a protective association between residential surrounding greenspace and anxiety and somatization among women in three different Spanish areas within two biogeographic regions. The reduction of anxiety and depression rates and the promotion of mental health in our societies are of prime importance, especially for women, who are more vulnerable to suffering from anxiety and depression. These findings, if replicated by other studies in other areas, could provide policymakers with the evidence base to implement interventions aiming at promoting mental health in our rapidly urbanizing societies. We recommend future studies gathering information on visits to and the time spent in different types of green spaces and relying on repeated measurements of exposure and outcome in a longitudinal frameworkwhile exploring the potential mechanisms underlying such an association.

## Figures and Tables

**Figure 1 ijerph-17-05670-f001:**
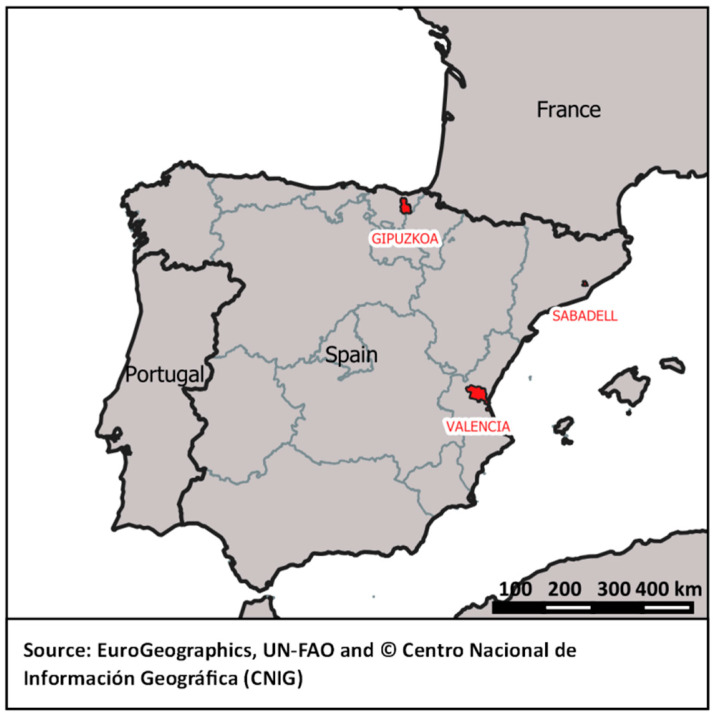
Location map representing the three study areas in Spain.

**Figure 2 ijerph-17-05670-f002:**
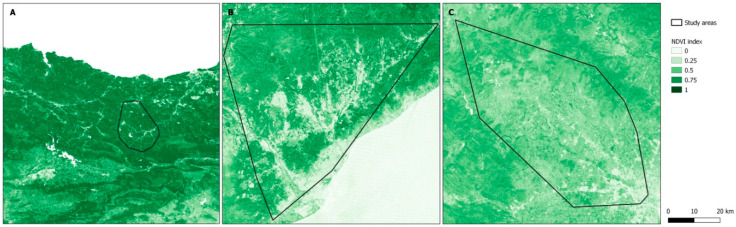
Landsat NDVI imagery and study areas. (**A**) Guipuzkoa (June 2010), (**B**) Valencia (May 2010), (**C**) Sabadell (May 2007).

**Table 1 ijerph-17-05670-t001:** Description of characteristics of the study participants separately by study area.

Covariates	Valencia	Sabadell	Gipuzkoa	All
**Nº of participants n (%)**	444 (37.9%)	475 (40.5%)	252 (21.5%)	1171 (100%)
**Age mean (SD)**	34.6 (4.1)	34.7 (4.1)	35.4 (3.3)	34.8 (4.0)
**Ethnicity n (%)**				
**White**	434 (97.7%)	450 (97.8%)	248 (98.8%)	1132 (98%)
**Others**	10 (2.3%)	10 (2.2%)	3 (1.2%)	23 (2%)
**Maternal education n (%)**				
**Primary school**	118 (26.6%)	111 (23.6%)	29 (11.5%)	258 (22.1%)
**Secondary school**	188 (42.3%)	208 (44.2%)	94 (37.3%)	490 (41.9%)
**University**	138 (31.1%)	152 (32.2%)	129 (51.2%)	419 (35.9%)
**Smoking n (%)**				
**Yes**	149 (34%)	132 (28%)	34 (15%)	315 (27.7%)
**No**	289 (66%)	340 (72%)	192 (85%)	821 (72.3%)
**Maternal alcohol consumption n (%)**				
**Yes**	46 (10.4%)	52 (11.5%)	14 (5.7%)	112 (9.8%)
**No**	395 (89.6%)	402 (88.5%)	232 (94.3%)	1029 (90.2%)
**Marital Status n (%)**				
**Married**	334 (85.4%)	365 (85.8%)	210 (93.3%)	909 (87.2%)
**Others**	57 (14.6%)	61 (14.2%)	15 (6.7%)	133 (12.8%)
**IQ mean (SD)**	10 (3.3)	10.5 (2.9)	9.5 (2.7)	10.1 (3.0)
**Neighborhood Socioeconomic Status ^(a)^ mean (SD)**	0.6 (0.2)	0.6 (0.2)	0.4 (0.2)	0.5 (0.2)

^(a)^ Urban Vulnerability Index.

**Table 2 ijerph-17-05670-t002:** Description of the average residential surrounding greenspace at delivery and the 4-year follow-up (NDVI), residential surrounding greenspace at the 4-year follow-up (NDVI), and psychopathological risk profile (SCL-90-R) separately by study area.

Greenspace Exposures and Mental Health	Valencia		Sabadell		Gipuzkoa		*p* Value ^(a)^
Variables							
Average residential surrounding greenspace delivery/4-year (NDVI)							
100 m buffer	0.24 (0.11, 0.60)		0.20 (0.11, 0.54)		0.44 (0.15, 0.81)		0.0001 **
300 m buffer	0.26 (0.15, 0.56)		0.24 (0.12, 0.56)		0.49 (0.22, 0.81)		0.0001 **
500 m buffer	0.27 (0.17, 0.54)		0.26 (0.12, 0.51)		0.54 (0.29, 0.81)		0.0001 **
Residential surrounding greenspace at 4-year (NDVI)							
100 m buffer	0.27 (0.09, 0.64)		0.20 (0.11, 0.53)		0.51 (0.15, 0.85)		0.0001 **
300 m buffer	0.27 (0.15, 0.56)		0.24 (0.12, 0.56)		0.52 (0.24, 0.82)		0.0001 **
500 m buffer	0.28 (0.16, 0.54)		0.26 (0.12, 0.51)		0.55 (0.30, 0.84)		0.0001 **
Psychopathological risk profile (the SCL-90-R)	Yes	No	Yes	No	Yes	No	
Global Severity Index (GSI)	6.9%	93.0%	10.5%	89.5%	6.7%	93.2%	0.08
Somatization	7.6%	92.3%	10.7%	89.3%	5.2%	94.8%	0.02 **
Obsessive-Compulsive	9.9%	90.1%	9.5%	90.5%	7.1%	92.9%	0.45
Interpersonal sensitivity	7.2%	92.8%	9.7%	90.3%	11.1%	88.89%	0.18
Depression	6.5%	93.5%	10.9%	89.0%	6.3%	93.6%	0.02 **
Anxiety	7.6%	92.3%	8%	92.0%	5.6%	94.4%	0.46
Hostility	9%	91%	9.3%	90.7%	7.9%	92.1%	0.83
Phobic anxiety	6.7%	93.2%	8%	92.0%	8.7%	91.3%	0.60
Paranoid ideation	5.9%	94.1%	10.1%	89.9%	7.5%	92.5%	0.05
Psychoticism	7.9%	92.1%	10.3%	89.7%	10.7%	89.3%	0.34

^(a)^ Kruskal–Wallis test (Residential surrounding greenspace), Chi-square test (SCL-90-R).*p* value < 0.05 **.

**Table 3 ijerph-17-05670-t003:** Adjusted and unadjusted logistic regression models for each buffer of the average of residential surrounding greenspace at delivery and at 4-year follow-up, and risk of each symptomatic dimension of the SCL-90-R. Odds Ratio (OR) and 95% confidence intervals (95% CI) for 1-IQR increase in each continuous indicator of residential surrounding greenspace.

	100 m Buffer	300 m Buffer	500 m Buffer
**Global Severity Index (GSI)**			
Unadjusted	0.91 (0.69, 1.20)	0.85 (0.64, 1.14)	0.85 (0.64, 1.14)
Adjusted ^(a)^	0.94 (0.68, 1.29)	0.87 (0.62, 1.22)	0.85 (0.60, 1.21)
**Somatization**			
Unadjusted	0.70 (0.53, 0.94) **	0.68 (0.51, 0.93) **	0.69 (0.51, 0.93) **
Adjusted	0.63 (0.44, 0.90) **	0.64 (0.43, 0.93) **	0.63 (0.43, 0.93) **
**Obsessive-Compulsive**			
Unadjusted	0.96 (0.77, 1.21)	0.86 (0.67, 1.12)	0.83 (0.64, 1.09)
Adjusted	0.98 (0.75, 1.27)	0.86 (0.64, 1.16)	0.83 (0.60, 1.13)
**Interpersonal sensitivity**			
Unadjusted	1.17 (0.95, 1.45)	1.18 (0.94, 1.49)	1.17 (0.92, 1.48)
Adjusted	1.20 (0.94, 1.54)	1.22 (0.93, 1.61)	1.21 (0.92, 1.61)
**Depression**			
Unadjusted	0.95 (0.70, 1.30)	0.87 (0.63, 1.20)	1.17 (0.92, 1.48)
Adjusted	0.99 (0.72, 1.37)	0.91 (0.64, 1.29)	0.87 (0.60, 1.24)
**Anxiety**			
Unadjusted	0.87 (0.67, 1.14)	0.78 (0.57, 1.05)	0.77 (0.57, 1.04)
Adjusted	0.84 (0.60, 1.17)	0.69 (0.47, 1.02)	0.67 (0.45, 0.99) **
**Hostility**			
Unadjusted	0.96 (0.76, 1.21)	0.92 (0.71, 1.19)	0.90 (0.69, 1.17)
Adjusted	1.02 (0.79, 1.33)	0.95 (0.71, 1.27)	0.92 (0.68, 1.24)
**Phobic Anxiety**			
Unadjusted	0.90 (0.70, 1.16)	0.94 (0.71, 1.23)	0.97 (0.74, 1.28)
Adjusted	0.96 (0.72, 1.29)	1.01 (0.73, 1.38)	1.04 (0.75, 1.43)
**Paranoid Ideation**			
Unadjusted	1.03 (0.78, 1.37)	1.02 (0.75, 1.39)	1.04 (0.75, 1.43)
Adjusted	1.06 (0.77, 1.45)	1.06 (0.75, 1.50)	1.08 (0.75, 1.55)
**Psychoticism**			
Unadjusted	1.04 (0.84, 1.29)	1.02 (0.81, 1.30)	1.02 (0.80, 1.30)
Adjusted	1.01 (0.78, 1.31)	0.98 (0.74, 1.31)	0.98 (0.73, 1.32)

^(a)^ Adjusted for age, smoking, urban vulnerability index, educational attainment and IQ. *p*-value < 0.05 **.

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
