# Peer review of "Residential Surrounding Greenspace and Mental Health in Three Spanish Areas"

_ijerph, 2020, doi:10.3390/ijerph17165670_

Round 1

Reviewer 1 Report

This is a very interesting and relevant research especially with regards to mental health. I have some specific comments: 

Abstract:

Line 29- 31 it will be good to clarify in that sentence that your study was to evaluate mental health in mothers within a birth cohort. 

Introduction:

The introduction is mostly about mental health statistics which is good but very little information is provided on studies looking at beneficial effects of green space. even though authours have mentioned that there are limited studies, it will be good if they can expand a little more on the studies they have mentioned (line 66 - 69). 

Line 52 - The sentence has grammatical error. Please correct the sentence. Perhaps say : Women are twice as likely to develop depression and anxiety than.....

Line 68 -69 - reword the sentence....say something like "however, evidence is limited on this aspect.  Also, please calrify "across different settings. Perhaps add another sentence on this aspect as I am not clear what you mean by "across different settings". 

Materials and methods: I note that pregnant women were enrolled in the study based on certain criteria. Was the mental health data completed at the same time as the satellite imagery? 

Line 95 - You said "data was collected...Please can you clarify what data was collected? The year of data collection was between was between 2003 to 2012, however, again what dat was collected during the different time points? What year did the initial cohort recruited the participants? I feel like more explanation is required here. 

Line 101 -109 - Was green space assessed at different time points? I see that in line 107 -108 you mention when these data was collected. Please can you re-write this paragraph so that its clear from the beginning that you collected the data at time of delivery and at 4 year followup. 

Line 111 - 126 - Was mental health assessed at birth as well as at 4 year follow-up? Please clarify. Do you also assess where they have lived during the 4 year period? 

Results: I note that these are women who had given birth and followedup at 4 year phase, was postnatal depression taken into account? This is a main confounding factor in such a birth cohort. 

Limitations

I feel that the biggest limitation in your study is not measuring postnatal depression which these mothers could have been effected with. It will be good to add this as a limitation to your study. 

Reviewer 2 Report

This article has a great job behind. However, some aspects it isn't clear. For these reason, it's necessary to explain and give more details in some paragraph.

Could be necessary a location map with the three location in Spain, maybe located in Europe, too. 

Is necessary explain the selection of participants, is to say, why pregnant women. Moreover, could be explain why this three location in Spain and not others. 

Moreover, the paragraph 2.2 is not very clear to understand. Are you analyze every space of 1.171 participants? 

The study population characteristics it is not possible to have it in the results section. A better option is created a Methodological section and inside this explain the study population, statistical analyses, sensitivity analyses, etc. 

In the section 3.3, would be interesting for researchers to make some maps with the buffers in the 3 study area. Is an option more visual. 

The conclusion is very short and does not include some important aspects covered in the article, such as the difference between the three study area. 

Reviewer 3 Report

Toda and co-workers are suggesting an interesting manuscript for publication that studied the associations of residential surrounding green space and mental state status in three diverse areas in Spain. The manuscript was a pleasure to read through. The science seemed sound and data interpretation appropriate. I did not detect any major errors or data over interpretation. The English was good.

I suggest that the manuscript can be accepted as it is with just one important revision:

The reference list need to be revised, as per journals policy. Many references in the list do not have appropriate source information. Numbers:

5, 7, 9, 38, 40. The most blatant example is reference 38.

Some references likely miss the journal: 13, 16, 17, 18, 20, 21, 23, 26, 27, 36,. Showing only DOI is not appropriate styling.

Please correct this.

Otherwise, congratulations!

Round 2

Reviewer 1 Report

Thank you for your responses. Please make sure that there are no grammatical errors or spelling mistakes. Other than that I am happy with your changes. 

Author Response

Dear Reviewer, 

We have checked the manuscript again and now, we are sure that there are no grammatical errors or spelling mistakes. Many thanks for helping us to improve our work.